# Interventions to Facilitate Return to Work after Stroke: A Systematic Review

**DOI:** 10.3390/ijerph20156469

**Published:** 2023-07-28

**Authors:** Gemma Pearce, Joan O’Donnell, Rebecca Pimentel, Elizabeth Blake, Lynette Mackenzie

**Affiliations:** 1School of Health Sciences, Faculty of Medicine and Health, The University of Sydney, Sydney, NSW 2006, Australia; 2Occupational Therapy Department, Concord Repatriation General Hospital, Concord, NSW 2139, Australia; 3Occupational Therapy Helping Children, Frenchs Forest, NSW 2086, Australia; 4Occupational Therapy Department, Royal North Shore Hospital, St Leonards, NSW 2065, Australia

**Keywords:** stroke, cerebrovascular accident, CVA, work, employment

## Abstract

Purpose: To gather knowledge about effective return to work interventions for survivors of stroke. Methods: A database search was conducted in MEDLINE, CINAHL, PsycINFO, Scopus, and Web of Science using keywords and medical subject headings. Studies were included if they met the following criteria: (i) studies published in English since the year 2000; (ii) adult patients aged 18–65 with a primary diagnosis of stroke; (iii) working pre-stroke; and (iv) intervention in which one of the primary outcomes is return to work. The methodological quality of included studies was assessed and the evidence synthesised. Results: Twelve studies were included, of which three were randomised controlled trials, four were retrospective studies, one was a cohort study, one was an explorative longitudinal study, one was a pre-post treatment observation study and two were pilot studies. The employment rate at follow-up ranged from 7% to 75.6%. Overall, there was limited published evidence regarding the effectiveness of interventions to promote return to work for this population, and it was unclear if return to pre-stroke work was the goal. Conclusion: A lack of large, controlled trials, variations in follow-up time and the definitions of return to work accounted for the large range of employment rates at follow-up. There is limited published high-quality evidence regarding the effectiveness of interventions to promote return to work in working-age survivors of stroke.

## 1. Introduction

Stroke has historically been viewed as a disease affecting mainly older people, however the incidence of younger people with stroke is steadily increasing [1]. Approximately 30% of stroke survivors are under the age of 65 [2], meaning many people will be affected by stroke in their income-earning years. Stroke survivors can experience long-term cognitive and physical impairments, which can pose a significant barrier to reintegration into the workplace [3,4,5]. Failure to return to work can result in increased mortality rates, poor general health, and poor mental health and psychological wellbeing, in addition to loss of self-esteem, social participation, fulfilment and income [6]. This not only affects the emotional wellbeing and financial stability of both an individual and their family but has a financial effect on society at large due to loss of productivity and reliance on government-funded income supplements [7]. Furthermore, the more time someone spends off work the less likely they are to ever return [6], so receiving rehabilitation that promotes return to work early in recovery from stroke is vital. However, rehabilitation efforts have not been designed to meet the needs of the young stroke survivor wishing to re-enter the workforce [8,9,10].

Qualitative research has shown that rehabilitation of younger stroke survivors does not sufficiently address the issue of return to work. Lock et al. [8] found from focus group discussions with stroke survivors and their supporters that many people consider the rehabilitation system to be a barrier to employment and perceived it to be insufficient in duration and in scope to prepare them for work. Medin, Barajas and Ekberg [9] conducted qualitative interviews with Swedish stroke survivors and similarly found that they perceived rehabilitation as focusing mainly on bodily functions and carrying out everyday activities rather than promoting return to work. Many participants reported that they felt frustrated at the lack of rehabilitation supporting returning to work, as that was a major personal goal. While qualitative studies are essential to understand the experience of returning to work by stroke survivors, they are unable to establish efficacy of interventions. 

Röding et al. [10] also found considerable consensus among participants regarding the lack of participation experienced in hospital and during rehabilitation following stroke. Participants reported feeling as though the rehabilitation they received was not age-adapted, and the needs of younger stroke survivors were not addressed by the surroundings and activities offered during rehabilitation. These studies indicate that rehabilitation efforts sseem to be inadequate in preparing stroke survivors for work after stroke.

It is evident that effective stroke rehabilitation needs to include intervention to promote return to work, however little is known about interventions that successfully achieve this. Donker-Cools et al. [11] conducted a systematic review which aimed to gather knowledge about effective return to work interventions for patients with acquired brain injury (ABI). They performed an extensive search for articles available in English, French, German and Dutch published between 2000 and 2015, searching six databases using a multitude of search terms that were developed with a clinical librarian. Despite their extensive search strategy, after screening and assessing articles according to their eligibility criteria, and reviewing the reference lists of relevant studies, only 12 articles were retained. Of these twelve studies, five comprised patients with ABI, five comprised patients with traumatic brain injury (TBI), one study involved a variety of neurological problems and only one study comprised stroke patients. After synthesising the results from all the included studies, Donker-Cools et al. [11] concluded that interventions containing work-directed components and education and coaching are effective regarding return to work after acquired brain injury. However, as only one of their included studies comprised stroke patients, it is unclear if this conclusion is applicable to stroke survivors.

Brouns et al. [12] conducted a systematic review that aimed to gather knowledge about effective return to work interventions following ischaemic stroke. Their search strategy was also extensive, covering six databases and including dedicated controlled vocabulary for search terms. They also consulted relevant prospective trial registers and the grey literature and completed backwards and forward reference searching. They did not have any limitations regarding the date of publication and included articles in English, French, German, Spanish and Dutch. Again, despite their extensive search strategy, Brouns et al. [12] found only two relevant studies; a retrospective study which was assessed to have low methodological quality, and a prospective study which they reported may have sufficient quality, however, due to the lack of randomisation, variability of follow-up, and restrictive patient selection criteria, advised caution is warranted when interpreting the results. The retrospective study reported on the effectiveness of an outpatient programme to promote return to work after ischaemic stroke, and found that following this intervention 30% of patients returned to work after three months. The prospective study evaluated the effect of intravenous thrombolytic therapy and found that patients who received this therapy were twice as likely to return to work as those who did not. Because of the limited number of relevant studies, they concluded that there is insufficient evidence regarding the effectiveness of interventions to promote return to work for this population.

Wei, Liu and Fong [13] similarly conducted a systematic review that aimed to identify the outcomes of return to work for stroke survivors after rehabilitation. Their search strategy was less extensive than the previous two systematic reviews described. They searched only three databases (MEDLINE, CINAHL and PubMed), and used relatively few search terms compared to the other mentioned studies, to find relevant articles published in English between 2004 and 2014. Wei, Liu and Fong [13] did not specify in their review whether multiple authors were responsible for identifying and evaluating studies and extracting data, so the risk of bias in this study was high. Despite their less exhaustive search, Wei, Liu and Fong [13] found 10 studies that met their inclusion criteria. However, only three of these studies introduced the rehabilitation programmes in which subjects had participated; many studies simply reported the hospital which their sample was recruited from without mention of the treatment that was received. The majority of included studies were either single cohort studies or single group trials where study participants were not compared with a control group, except for the single relevant RCT that was found. They concluded that there is limited evidence to support vocational rehabilitation as an intervention to promote return to work after stroke.

More recent reviews by Ashley et al. [14], O’Keefe et al. [15], Green et al. [16], Proffett et al. [17] and La Torre [18] agreed that return to work for people following stroke was associated with a wide range of factors that could assist in the process, such as support from health professionals and family, socioeconomic factors, being younger, functional capacity and type of work undertaken. Several reviews included qualitative studies exploring the experiences of return to work for participants following stroke. However, few studies examined or described an intervention that was implemented to facilitate return to work and measured actual work outcomes.

The lack of controlled trials investigating return to work interventions post-stroke means it is difficult to determine if the effect observed is due to the intervention or merely the effect of the time course. It has also been difficult to determine a baseline rate of return to work in this population due to the use of different definitions of ‘return to work’ and ‘stroke’, and differences in follow-up times in the published literature. Duong et al. [19] recently conducted a systematic review to examine operational definitions of ‘return to work’ in post-stroke literature and provide more precise estimates of return to work through meta-analysis. They found 55 studies that met their inclusion criteria, 33 of which reported an operational definition of return to work. The majority of studies considered successful return to work to be resumption of either full-time or part-time work. Some studies included being a student, some included homemaking, some included self-employment, casual work, and/or temporary work and some studies explicitly excluded one or all of these. Some studies used cessation of sickness leave/benefits as a measure of return to work; however, as identified by Duong et al. [19], this may lead to an overestimation of return to work rates, as a person may have ceased receiving benefits if they have surpassed the maximum duration, but may not have returned to work. They proposed that the following operational definition be used in all future research concerning return to work after stroke to allow greater accuracy when making comparisons between studies: “the resumption of any paid work full-time or part-time, inclusive of self-employment, in a regular or modified capacity, for a committed average number of hours of work each week” [14] (p. 1149). Duong et al. [19] found that the follow-up time in the included studies ranged from less than one month to 18.8 years. As follow-up time is a significant determiner of return to work, this was identified as another area for improvement in future research; i.e., a standardized follow-up period among studies will make it easier to compare their reported return to work rates. Duong et al. [19] suggested that this could be six months, one year and two years post-stroke as these were most commonly used among the identified studies. They found that across all included studies, the rate of return to work ranged from 4% to 90%. They also found there was a similarly large range in reported return to work rates for stroke survivors who participated in rehabilitation (7.3% to 90.9%). Duong et al. [19] pooled the studies with the same follow-up times to provide more precise summary estimates of return to work: 53.2% at six months, 55.7% at one year and 67.4% at two years. These estimates will be useful as benchmarks to assess the effectiveness of interventions striving to improve this outcome when there is no control group, as is the case with much of the return to work literature.

As discussed above, previous systematic reviews conducted on this topic have had limited success identifying studies that investigated interventions to promote return to work after stroke [11,12,17], and those that have been able to identify a larger number of studies have been conducted with a high risk of bias [13]. Furthermore, the reviews by Donker-Cools et al. [11] and Wei, Liu and Fong [13] only included studies published up to the years 2015 and 2014, respectively. Consequently, there is a lack of up-to-date information about effective return to work interventions for survivors of stroke. We decided that a systematic review was needed that focused only on stroke survivors and encompassed literature from 2000 to 2022 in order to assist clinicians to provide services that could address return to work. The aim of this study was, therefore, to gather knowledge about effective return to work interventions for survivors of stroke in a systematic way. The research question was, “What are effective return to work interventions for survivors of stroke?”.

## 2. Methods

This research followed the guidelines laid out in the PRISMA-P 2015 statement for reporting systematic reviews [20]. These guidelines involve applying the research question to the participants, interventions, comparators and outcomes (PICO). As implied in the research question, the participants were stroke survivors of working age, the interventions was any intervention used to assist return to work, the comparator was usual care for controls who did not receive the intervention, and outcomes were any outcome measured by the studies included in the systematic review.

### 2.1. Search Strategy

To collect research about interventions that focus on return to work after stroke, the following databases were searched: MEDLINE, CINAHL, PsycINFO, Scopus and Web of Science. The search terms were formulated in PubMed in collaboration with a health sciences librarian and adapted to make them applicable for the other databases. Both keywords and medical subject headings (MeSH) terms were used. The searches were limited to articles available in English, and published since 2000. See Appendix A for full details of the search strategies and search terms. A search of the reference lists of included/other relevant studies was also undertaken. The original database search was conducted on 28 September 2019 and re-conducted on 30 December 2022.

### 2.2. Selection Criteria

Studies retrieved following the search were identified against the inclusion and exclusion criteria. The inclusion criteria were: (i) studies published in English; (ii) studies published since the year 2000; (iii) the study population comprised patients aged 18–65 with a primary diagnosis of stroke, or where studies included multiple diagnostic groups, the stroke group results were reported separately; and (iv) study participants had participated in rehabilitation or received an intervention in which one of the primary outcomes was return to work. Return to work in this systematic review is characterised as “the resumption of any paid work full-time or part-time, inclusive of self-employment, in a regular or modified capacity, for a committed average number of hours of work each week”, as suggested by Duong et al. [19] (p. 1149). Studies were included with the following designs: randomized and non-randomized controlled trials; case control studies; cohort studies; retrospective studies; and longitudinal studies. Previous systematic reviews and qualitative studies were excluded.

### 2.3. Study Selection

Studies retrieved by the search were initially assessed for relevance independently by two authors on the basis of title. One of the authors then reviewed the abstract and/or full-text of the retained studies against the selection criteria above. A record of rejected studies and the reasons for rejection were documented and a second author checked and verified these decisions.

### 2.4. Data Extraction and Analysis

Data were extracted and methodological quality was assessed by one author and verified by a second author. Data were extracted from included studies using a data extraction form that included information on geographic location, study design, sample size, and sample demographics including gender, age and level of education. It also included information on the intervention, controls, treatment duration, follow-up time, and vocational status pre-stroke, pre-intervention, at discharge and at follow-up. To evaluate the quality of included studies, the Downs and Black checklist was used [21]. This checklist has been shown to be a reliable and valid measure of quality for both randomized and non-randomized studies [21]. It consists of 4 domains; reporting (10 items), external validity (3 items), internal validity (13 items), and power (1 item). As the interpretation of question 27 on the checklist (the power domain) is unclear, the below provision was used instead: a score of zero was assigned if no power calculation was provided, a score of 3 was assigned if a power calculation was provided but the importance of the difference between groups is unclear, and a score of 5 was assigned if the difference between the groups was clearly defined as a clinically important difference. A second author checked the extracted data and appraisal against the full-text reports of included studies.

## 3. Results

### 3.1. Study Selection

Figure 1 shows the flowchart of the study selection process. The database search yielded 1031 records. An additional eight citations were identified after reviewing the reference lists of the included articles and three review articles that were retrieved by the search. After removing duplicates, the remaining 544 articles were screened for eligibility on the basis of title independently by two authors. A total of 42 articles were retrieved for full text review, of which 7 met the inclusion criteria. The updated search followed the same procedures and revealed an additional four articles published since the original search which met the inclusion criteria. The most common reasons for exclusion were that stroke group results were not reported separately, or that the studies did not involve an intervention. Documentation of rejected studies and the reasons for rejection are available on request.

### 3.2. Study Characteristics

The characteristics of the included studies are presented in Table 1. In the studies included in our systematic review, four are retrospective studies [22,23,24,25], three were randomised controlled trials [26,27,28,29,30], one is a cohort study [31], one is an exploratory longitudinal study [32], one is a pre-post treatment observation study [33] and two were pilot studies [34,35,36]. Two of the studies were conducted in the USA, two in Singapore, two in Sweden, one in South Africa, one in France, one in the UK, one in Denmark and two studies that included data from several different countries (Norway, China, USA, Russia, Palestine, Israel and Sweden; and Australia, New Zealand, United Kingdom, Malaysia and Singapore). Three interventions were presented across two articles each—the AVERT programme [26,27], ESSVR programme [28,29] and RE-WORK programme [34,35].
Figure 1PRISMA flow diagram [24].
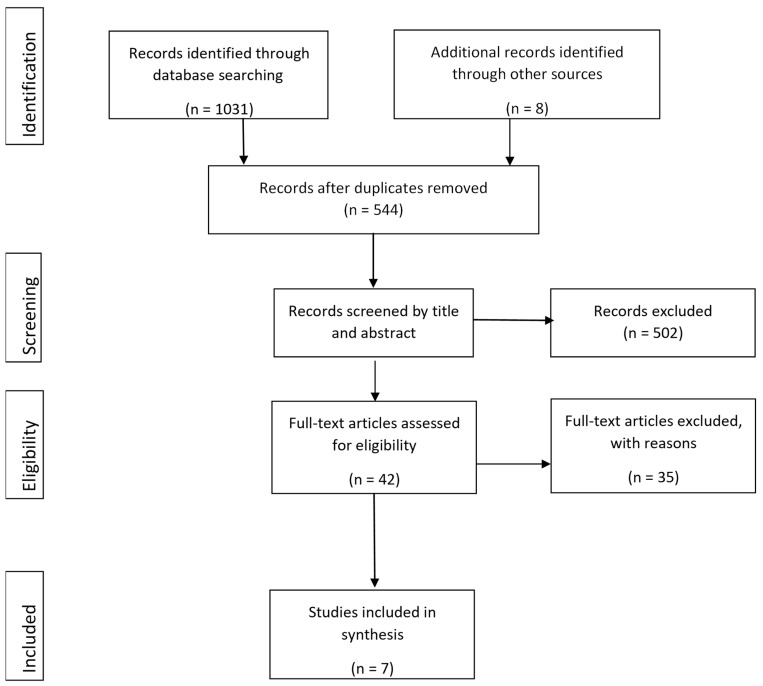


### 3.3. Methodological Quality Assessment

The methodological quality of the included articles was assessed using the Downs and Black checklist [21]. The scores of included articles are presented in Table 2. Overall, the quality of the included studies was low-moderate. The reporting domain, which reflects whether the information provided in the paper is sufficient to allow the reader to form an unbiased assessment of the study findings, was found to have the highest quality of all four domains, with a mean score of 5.72 (57% of the total score). The power domain was found to have low quality in all of the studies, as none of the included studies provided a power calculation.

### 3.4. Study Population

The characteristics of the study population are also presented in Table 1. Stroke severity was reported to be severe in one of the studies [32], and another study identified the proportion of participants with motor sequelae (80.4%), sensory impairment (48.2%), cognitive impairment (35.7%), language impairment (37.5%) and visual impairment (21.4%) [23]. The remainder of the included studies did not specify stroke severity or sequelae of participants. The total number of subjects included in all included studies was 1183. Sample sizes ranged from 29 [22] to 230 [32]. Participant gender was identified in all included studies except one [22]. The mean age of participants was reported in seven studies and ranged from 43.9 [34,35] to 65.75 [24]. Median age was reported in three studies and ranged from 44 [25] to 56 [26,27]. One study did not report the mean or median age of participants [22].

### 3.5. Follow-up

The follow-up time in the included studies ranged from 4.6 weeks [30] to 3 years [23,31]. Five programmes included follow-up at one year [26,27,28,29,31,32,33]. Two studies did not report a follow-up time [22,25].

### 3.6. Interventions

Five programmes did not report vocational status prior to stroke [22,24,26,27,28,29,30] so it was difficult to determine the impact of an intervention. Summaries of the interventions described in the included studies are provided in Table 3. Two studies did not report the details of the intervention/rehabilitation received [23,32]. Of those programmes that were described, most were individually tailored and consisted of a range of work-related activities such as multidisciplinary assessment (including by occupational therapists, psychologists, physiotherapists and social workers), identifying barriers to work, workplace evaluation, work trials and grading of work activities, liaising with employers, teaching work skills and self-management and counselling. The duration of the interventions was reported in four of the included studies, ranging from 10 h [28,29] to 19 months [34,35].

Some early interventions took place in rehabilitation settings following a stroke and measured return to work as an outcome of the programme, so were included in this systematic review. These included early mobilisation and functional activities [26,27], constraint induced movement therapy [24] and balance and visual therapy [36].

### 3.7. Outcomes

Return to work was the primary or secondary outcome measure in all studies. There was little consensus on the definition of return to work between studies; for example, studies defined successful return to work as participants who were competitively employed, employed in a modified job, or participating in an educational programme [33], as working or studying for at least 10 h/week [31], or at least had one hour of paid employment [26]. Data on return to work were obtained through retrospective review of archival records [22,33], questionnaires/surveys either posted or conducted via telephone [23,24,29,30], participant medical records or personal communication with the study authors [31], or semi-structured interviews [26,32]. The pre-stroke employment rate ranged from 48% [24] to 100% [24,30,33]. The employment rate at follow-up ranged from 7% [31] to 75.6% [33]. Table 3 presents the employment rates reported in the included studies.

## 4. Discussion

The aim of this systematic review was to gather knowledge about effective return to work interventions for survivors of stroke which might not have been examined by previous systematic reviews [11,12,13,14,15,16,17,18]. Of note is that some studies included relatively young participants—for instance, Adams [33] and Mohamad [25] reported participants were as young as 38. These participants had years of work ahead of them and would need different interventions to maintain their work productivity compared to older participants who may have had options for retirement. Similar to Donker-Cools et al. [11] and Wei, Liu and Fong [13], only three randomised controlled trials that investigated the effectiveness of an intervention to promote return to work after stroke were identified in this systematic review [26,27,28,29,30]. Ntsiea’s [30] was assessed as being the highest-quality RCT, and this study demonstrated that an individual-specific workplace intervention programme involving a plan to overcome identified barriers developed by therapist, patient and employer was effective regarding return to work. The intervention included evaluation and adaptation of work tasks, hours and environment, and included vocational counselling consisting of coaching and advice on coping strategies [30]; however, the treatment duration and vocational status at admission and discharge were not reported. When compared to the control group who received usual stroke care, participants in the intervention group had 5.2 greater odds of returning to work six months after participating in the workplace intervention programme [30]. One RCT was a sub-study of a RCT but had a larger sample of 376 participants from 5 countries; however, there was no significant difference between the intervention and control group in the odds of returning to work and no significance difference between countries [26]. The other RCT was a feasibility RCT [29] consisting of only 46 participants with no significant differences in return to work between intervention and control groups reported.

The study by Adams et al. [33] had the greatest rate of return to work at follow-up out of all of the identified studies, which may suggest that their reported intervention (a mixture of group and individual therapy activities teaching compensatory strategies, arranging environmental supports, providing counselling and education) is also an effective intervention to promote return to work after stroke. However, the definition of ‘work’ used by Adams et al. [33] was much broader than that of any of the other included studies as it included competitive employment, modified employment, participation in an educational programme, homemaking, supported employment and volunteering, whereas the majority of the other included studies would have only included competitive or modified employment. When looking only at those who were in competitive or modified employment, the rate of successful return to work at follow-up became 48.9%.

As the majority of included studies did not have a control/comparison group, it is difficult to know what effect, if any, was due to the intervention and what was due to the passing of time, as an increase in follow-up time is associated with greater rates of return to work [14]. Adams et al. [33], Hofgren et al. [31] and Langhammer et al. [32] all included a follow-up time of one year, which means we can compare the rates of return to work found in these studies to the pooled one-year summary estimates of return to work found by Duong et al. [14]. Duong et al. [14] found the estimate of successful return to work one year after stroke to be 55.7% when pooling all of the return-to-work data available in their review. The return-to-work rates one-year post-stroke/intervention cessation were found by Adams et al. [33], Hofgren et al. [31] and Langhammer et al. [32] to be 48.9% (using the definition or return to work noted above), 7% and 20% respectively. This would indicate that the interventions did not increase the rate of return to work relative to the baseline rate. However, it is important to note that this pooled estimate was developed using results from studies with very different populations and definitions of ‘return to work’, and may have also included studies in which participants had attended some kind of rehabilitation, so this cannot be interpreted as if it were a control group.

The results of the quality evaluation suggest that studies investigating return to work outcomes after stroke rehabilitation/intervention had on average low-moderate methodological quality. Only half of the studies scored 50% or more of the quality indicators. Furthermore, none of the 12 studies reported on any power calculations relevant to their study, which seriously undermines the quality of the findings. Due to the lack of controlled trials, and low-moderate methodological quality of included studies, there is inconclusive evidence to support the effectiveness of interventions to promote return to work after stroke. It is evident that further high-quality research needs to be conducted, with a focus on controlled trials of return to work interventions for stoke survivors. It is also of note that while this systematic review included studies from various countries, the studies were conducted in economically developed countries that offer social benefits for people with disabilities. It is likely that return to work rates will vary depending on the presence/absence of government-funded income supports [14], so future research should also aim to include a more diverse range of study populations.

Another important consideration that has not been discussed in the literature is the assumption that a return to new employment is considered successful ‘return to work’. Doucet et al.’s [23] was the only included study that differentiated between the rate of people who returned to their previous work position (50% of the people who returned to work) and those who changed jobs (the remaining 50%). There could be effects on a person’s motivation and satisfaction if they are forced to find a new, potentially less personally meaningful or a less senior job. If a person has ‘returned to work’ but not to a job that they enjoy doing or find fulfilling, this may not be considered an indicator of success of rehabilitation. While it may sometimes be necessary for a survivor of stroke to find new employment if they no longer meet the inherent requirements of their previous role, interventions that aim to increase return to work after stroke should first aim to increase return to pre-stroke work. Thus, future research on this topic should aim to include return to pre-stroke work (even if in a modified capacity) as a separate outcome.

### 4.1. Limitations

The identified studies included highly heterogenous designs, outcome measures, follow-up times and definitions of ‘work’, meaning comparison between them is difficult. Additionally, a lack of good-quality controlled trials means it is difficult to determine if the effect seen is due to the intervention or an increase in follow-up time. This systematic review only included studies available in English, and there may be additional relevant studies published in other languages that were not included in this systematic review.

### 4.2. Conclusions and Directions for Future Research

Return to work is an important outcome of rehabilitation for working-age survivors of stroke; however, there is limited published evidence regarding the effectiveness of interventions to promote return to work in this population. Future research should focus on conducting controlled studies of interventions, while ensuring that definitions of return to work and follow-up times remain consistent with those suggested by Duong et al. [14].



**Implications for Rehabilitation:**

The incidence of stroke in people under the age of 65 is rising, so returning to work is an important outcome of rehabilitation.Rehabilitation efforts have not been designed to meet the needs of younger stroke survivors wishing to return to work, and there is limited available evidence for the efficacy of interventions to promote return to work in this population.Further research needs to be conducted and appropriate interventions implemented to ensure that stroke survivors receive the necessary assistance to transition back to work.


## Figures and Tables

**Table 1 ijerph-20-06469-t001:** Characteristics of included studies and samples.

Study	Country	Study Design	Sample Size	Sample Gender	Sample Age (Years)	Sample Education	*n*	Time Since Stroke Onset/Follow-Up Period Post-Stroke
Adams et al. [33]	USA	Pre-post-treatment observation study	127	77 male, 50 female	Median: 48 25th percentile: 38 (IQR 38–54) 75th percentile: 54	<High school diploma High school diploma or equivalent Some college College degree	9 35 29 54	12 months
Bin Zainal et al. [25]	Singapore	Retrospective study	50	37 male, 23 female	Median 44 (IQR 38–48)	Not reported		Not reported
Chan [22]	Singapore	Retrospective study	29	Not reported	Not reported	Not reported		Not reported
Doucet et al. [23]	France	Retrospective study	56	35 male, 21 female	Mean 48.3 ± 10.1	Junior high Vocational High school	13 22 21	≥3 years
Grant et al. [28] Radford et al. [29]	UK	Feasibility RCT	46	36 male, 10 female	Mean 58.3 ± 12.7 (intervention) Mean 53.8 ± 12.6 (control)	Not reported		3, 6 and 12 months
Hofgren et al. [31]	Sweden	Cohort study	58	44 male, 14 female	Mean 52 ± 7.9	Not reported		1 year and 3 years
Johansson et al. [34] Öst Nilsson [35]	Sweden	Pilot study	10	6 male, 4 female	Mean 43.9	Not reported		3 months and 9 months
Langhammer et al. [32]	Norway, China, USA, Russia, Palestine, Israel, Sweden	Exploratory longitudinal study	230	154 male, 76 female	Mean 58.5	0–6 years of education 7–12 years of education 13+ years of education	32 93 105	6 months and 12 months
Mennemeyer et al. [24]	USA	Retrospective study	121	69 male, 52 female	Mean 65.75	High school College or higher	49 51	24 ± 16 months
Ntsiea et al. [30]	South Africa	RCT	80	41 male, 39 female	Mean 45 ± 8.7	Degree >Grade 12 Grade 12 Grade 11 <Grade 7	6 16 24 27 7	4.6 ± 1.8 weeks
Schow et al. [36]	Denmark	Pilot study	29	16 male, 13 female	Mean 56.86 ± 7.5	Not reported		Time since stroke 3–36 months. 6 month follow-up.

**Table 2 ijerph-20-06469-t002:** Scores of the Downs and Black checklist for each study.

			Internal Validity (13)			
Study	Reporting (11)	External Validity (3)	Bias (7)	Confounding (6)	Power (5)	Total Score (32)
Adams et al. [33]	7	3	4	4	0	18
Bin Zainal, et al. [25]	6	2	3	2	0	13
Cain et al. [26] Langhorne et al. [27]	5	3	5	4	0	17
Chan [22]	5	2	3	0	0	10
Doucet et al. [23]	7	1	4	4	0	16
Grant et al. [28] Radford et al. [29]	5	3	5	3	0	16
Hofgren et al. [31]	8	1	4	5	0	18
Johansson et al. [34] Öst Nilsson [35]	6	0	4	3	0	13
Langhammer et al. [32]	6	0	4	2	0	12
Mennemeyer et al. [24]	8	1	3	1	0	13
Ntsiea et al. [30]	0	2	3	4	0	19
Schow et al. [36]	4	2	2	1	0	9

**Table 3 ijerph-20-06469-t003:** Interventions and primary findings of included studies.

Study	Intervention Summary	Treatment Duration	Vocational Status, n (%)			
			Pre-Stroke	Admission	Discharge	Follow-Up
Adams et al. [33]	Mixture of group and individual therapy activities teaching compensatory strategies, arranging environmental supports, providing counselling and education.	Median: 113 days 25th percentile: 78 days 75th percentile: 163 days	90 (100%) employed	Productive: 4 (4.4%) i. Competitive employment: 0 (0%) ii. Modified employment: 2 (2.2%) iii. Educational programme: 1(1.1%) iv. Homemaker: 0 (0%) v. Supported employment: 1 (1.1%) vi. Volunteer: 0 (0%) Non-productive: 86 (95.6%)	Productive: 73 (81.1%) i. Competitive employment: 33 (36.7%) ii. Modified employment: 15 (16.7%) iii. Educational programme: 2 (2.2%) iv. Homemaker: (2.2%) v. Supported employment: 2 (2.2%) vi. Volunteer: 19 (21.1%) Non-productive: 17 (18.9%)	Productive: 68 (75.6%) i. Competitive employment: 35 (38.9%) ii. Modified employment: 9 (10%) iii. Educational programme: 3 (3.3%) iv. Homemaker: 9 (10%) v. Supported employment: 0 (0%) vi. Volunteer: 12 (13.3%) Non-productive: 22 (24.4%)
Bin Zainal et al. [25]	Transition to employment (TTE) vocational rehabilitation programme. Individualised services including physical rehabilitation, psychosocial, employment, and caregiver support. Assessment of the home and work environments, assistive technology, work task simulation, job accommodations and modifications.	Median of 10 months.	Not reported	Not reported	44 returned to work. 6 did not return to work	Not reported
Cain et al. [26] Langhorne et al. [27]	Very early mobilisation (AVERT) as an in-patient within 24 h assistance with functional tasks, sitting on edge of the bed, standing up, sitting out of bed and walking.	Not reported	64% <65 years old were working before stroke.	Not reported	Not reported	42% had returned to work at 3 months. Of those working at 3 months, 149 (90%) were still employed at 12 months. At 12 months, 221 (59%) had returned to work.
Chan [22]	Client interviewed by a work placement officer and assessed by OT, psychologist and social worker. Participation in job trials and job placements, upgrading skills course if necessary.	Not reported	Not reported	Not reported	Not reported	Employed: 16 (55%) i. Open employed: 10 (34%) ii. Sheltered workshop: 6 (21%)
Doucet et al. [23]	Patients attended a French Centre for Physical and Rehabilitation Medicine (CPRM) after a first stroke. Intervention not described.	195.7 ± 162.5 days (median: 145.5 days)	56 (100%)	Not reported	Not reported	18 (32.1%) Of those who returned to work, 9 (50%) returned to the same work position, and 9 (50%) changed jobs.
Grant et al. [28] Radford et al. [29]	Individually tailored early vocational rehabilitation with an occupational therapist, involving work preparation, discussing work options, timing of the return to work and the hours they felt able to manage. Teaching pacing and fatigue management. Applying for voluntary work, visits to Job Centre Plus, and looking for education and retraining opportunities. Use of routines and timekeeping, patient contact with workplace, detailed job analysis, and identifying potential problems and solutions. Return to work process including planning, monitoring and grading, and maintenance. Work site visits. Confidence building, behavioural problems, dealing with others, and teaching relaxation techniques. Mobility training, driving assessment, using public transport. Memory and executive functioning training. Referral and consultation with specialist services.	Mean of 10 (SD 7, range 1–25) intervention sessions (approx. one hour per session).	Not reported	2 in full-time education 7 were volunteering 69.6% of intervention group (16/23) and 73.9% of control group (17/23) were working full-time, and 1 (1/23; 4.3%) and 3 (3/23; 13%), respectively, were working part-time. Ten of intervention group (21%) were self-employed and working full-time baseline, six (6/23;26.1%) of control group were self-employed.	Not reported	31; 67.4% returned to work at some point following stroke; 9 (19.6%) did not RTW during up to 12-month follow-up (intervention = 4; control = 5). (8/24) 52% who were employed full-time at stroke onset resumed full-time work at 12 months post-stroke.
Hofgren et al. [31]	Patients received 9 h of individually tailored training (from an OT and PT) per week for 3 weeks either at home or at the clinic after discharge from a rehabilitation ward. Focus on activities in their natural context, content including personal care, shopping and leisure.	3 weeks	55 (95%) (included 4 people who were unemployed at the time of stroke but looking for work classified as ‘employed’ by study definition).	Not reported	Not reported	1 year: 4 (7%) 3 years: 11 (20%)
Johansson et al. [34] Öst Nilsson [35]	Preparation phase: resources and hindrances for RTW are mapped and a plan for work trial is made. Work trial phase: training at the workplace. Occupational therapist co-ordinates stakeholders and work visits.	4.5–19 months	Not reported	Not reported	Not reported	At 9 months: 3 still working. 7 returned to paid employment to some extent.
Langhammer et al. [32]	Specialised rehabilitation for people whose needs are beyond the scope of local rehabilitation services. Intervention not described.	Mean: 49 days	123 (53.4%)	Not reported	Not reported	6 months: 36 (18%) 12 months: 37 (20%)
Mennemeyer et al. [24]	Constraint induced movement therapy.	Not specified.	56 (48%)	25 (22%)	Not reported	29 (25%)
Ntsiea et al. [30]	Workplace intervention programme.	Not reported	80 (100%)	Not reported	Not reported	At 3 months: Intervention: 27% Control: 2% 6 months: Intervention: 60% Control: 20%
Schow et al. [36]	Intensive balance and visual therapy in groups of six or seven, by physiotherapist and optometrist, and home exercises.	Four months	24% in some sort of employment, 3.3% working full time.			At 6 months 60% in some form of employment and 27.6% working full time.

## Data Availability

Not applicable.

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
