# Peer review of "Interventions to Facilitate Return to Work after Stroke: A Systematic Review"

_ijerph, 2023, doi:10.3390/ijerph20156469_

Round 1

Reviewer 1 Report

This study investigated the literature on returning to work for young stroke survivors, which had not been progressed much in the past. As stroke, which was considered a geriatric disease, occurs frequently in young people, it is a very important and interesting study.

As a study that followed the procedures of detailed review and systematic review of the existing literature, it was judged to be of high clinical significance and was judged as eligible for publication.

It would be more meaningful if you emphasized the importance of return-to-work interventions for young stroke survivors in the discussion.

There are many parts where spaces are not reflected when viewing PDF. Please check carefully before publishing.; Thereis(abstract), animportant, theefficacy, notsufficiently, seemto(introduction)

Author Response

Thank you for the positive comments about the paper.

We have added the following sentences to the discussion relating to younger participants:

"Of note is that some studies included relatively young participants – for instance, Adams [33] and Mohamad [25] where participants were as young as 38. These participants have several years of work ahead of them and would need different interventions to maintain their work productivity compared to older participants who may have options for retirement."

The spacing issues appear to be a product of the template used for the article by IJERPH. I have adapted the examples given by the reviewer.

Reviewer 2 Report

Thank you very much for the opportunity to review this paper.

The manuscript addresses an important question of effective return to work interventions for survivors of stroke. The  database search, which was conducted in MEDLINE, CINAHL, PsycINFO, Scopus, and Web of Science aimed to find interventions in which one of the primary outcomes was return to work. The authors searched studies that met inclusion criteria, like published in English since 2000, with sample of adult patients aged between 18-65 with a primary diagnosis of stroke, working pre-stroke.

The article has been written very clearly.  The manuscript structure, flow and writing do not need improving. The language of the paper meets the standards of academic English used in scientific journals. I found only one bunch of words: ”interventionprogram”, in Table 2, page 13.

Introduction
In my opinion, the search is well justified in Introduction. The literature chosen is adequate.

The Authors clearly emphasized the need of such search.

Method
The method used is very well described. The criteria of interventions inclusion are good.

Any additional tables or figures are not necessary.

Results and conclusions

12 studies were included. The employment rate at follow-up ranged from 7% to 75.6%. The results are well described. The interpretation of results and conclusions are supported by the data.

The Authors drew logical conclusions and emphasized that there is lack of large controlled trials, variations in follow-up time and the definitions of return to work accounted for the large range of employment rates at follow-up. They underline that there is limited published high quality evidence regarding the effectiveness of interventions to promote return to work in working-age survivors of stroke. I believe the conclusion about the lack of publications describing interventions for return to work after stroke is  very important one, encouraging both such interventions and publications about their results.

Authors underline important implications for rehabilitation, too, like the fact that incidence of stroke in people under the age of 65 is rising, so returning to work is an important outcome of rehabilitation, or that rehabilitation efforts have not been designed to meet the needs of younger stroke survivors wishing to return to work.

I believe this article may contribute to the existing literature.

Personally, I find the paper interesting, important and well written.

Thank you.

Author Response

Thank you for the positive review of our paper.

In response to "I found only one bunch of words: ”interventionprogram”, in Table 2, page 13." - these are related to the conversion of the document to the IJERPH template.  I have tried to space as many of these examples as possible.

I have also updated the Table numbers to be consistent with the presentation order of the Tables.

Author Response

Thank you for the review which has improved the paper.  See our response to the suggestions made below.

Page 1: Provide a statement or two on the general findings from the systematic reviews. Concerns about the quality of sources are important, though this can be complemented with an indication of what the general theme/pattern of results is from the assessment of the 12 studies included in this review.

These two statements were added to the abstract on page 1" "Overall, there was limited published evidence regarding the effectiveness of interventions to promote return to work for this population, and it was unclear if return to pre-stroke work was the goal."

Page 2: The review of the literature here features some qualitative studies. It might help to clarify here how the efficacy of stroke rehabilitation systems was assessed (if they were) from these qualitative studies. 

Efficacy cannot be established using qualitative studies that are focused on the experience of return to work by stroke survivors. 

This sentence added to the section on qualitative studies:

"Whilst qualitative studies are essential to understand the experience of returning to work by stroke survivors, they are unable to establish efficacy of interventions."

Page 2: It seems from previous studies and reviews (such as the Donker-Cools et al and later, Ashley et al, Duong et al. papers) that there is a general lack of studies examining the efficacy of stroke rehabilitation. It would be helpful at this point to propose a stronger justification for why then, the present study presents a follow-up systematic review of these interventions when it is known that studies are generally lacking in this area.

Pages 2-3: A general suggestion may be to strongly justify the current systematic review more clearly – why, for one, expand on the date range of published studies (2014-2015) – and more importantly, to highlight the added stringency of criteria that more strongly justifies the selection of a modest 12 studies for the present systematic review even if the date range was expanded to include studies published after 2000. 

This sentence was added in the justification:

"We decided that a definitive review was needed that focused only on stroke survivors and encompassed literature from 2000 to 2022 in order to assist clinicians to provide services that could address return to work."

Page 5, 9: It seems concerning that none of the included studies did not report the power calculation, and that not all included studies meet the ‘passing’ mark of scoring more than 50% (16 per 32) of the Downs and Black checklist. A suggestion to ‘reframe’ the current study might to be consider this as a critical evaluation of the quality of existing studies examining stroke rehabilitation programs.

These sentences were added to the quality section of the discussion:

"Only half of the studies scored 50% or more of the quality indicators. Furthermore, none of the 12 studies reported on any power calculations relevant to their study, which seriously undermines the quality of the findings. 
rehabilitation programs."

Page 13: The Ntsiea et al. [30] article entry seems to provide minimal details on the exact features and qualities of the stroke rehabilitation program. Many of the key details typical in the other included studies are presented, but not for this study. This raises some concerns about its inclusion in the present systematic review, despite it being argued to be a high-quality RCT intervention in the discussion section. 

All studies that met the inclusion criteria were included in the review despite their qualities and characteristics which have been reported. 

A sentence has been added in the description of the Ntseia study:

"however, the treatment duration, and vocational status at admission and discharge were not reported."

Page 14: Similar to an initial comment, it would be informative to detail the broader themes/findings from the review. As is, the outcomes seem varied and much too generic to indicate commonalities in findings across the different rehabilitation programs included and compared in the current work.

We can only report what the findings of the review reveal, and the findings are varied and may not have enough detail to identify specific commonalities across the interventions used since many were not detailed in their reporting. This has been made clear in the review.

Page 15: As with the initial comment, it might be preferred to frame this study as a critical evaluation of the quality of stroke rehabilitation interventions, rather than a systematic review of the findings/outcomes of such rehabilitation programs. The reporting seems to reflect more of the former than the latter.

We did not set out to review stroke rehabilitation in general, but specifically the outcomes of return to work for stroke survivors. Quality of the studies is part of a systematic review and these findings have been reported. We have not commented on stroke rehabilitation in general.

Round 2

Reviewer 3 Report

Thank you for the quick and detailed response to my earlier comments. This version of the manuscript looks good as is.

Author Response

It appears that the round 2 review comment is that nothing is required to be done.

Comment was: "Thank you for the quick and detailed response to my earlier comments. This version of the manuscript looks good as is."

You can use the previous manuscript submitted.

Lynette